# Concentrations of Major and Trace Elements within the Snowpack of Tyumen, Russia

**Dmitriy Moskovchenko *** , **Roman Pozhitkov, Aleksandr Zakharchenko and Aleksandr Tigeev**

Tyumen Scientific Centre, Siberian Branch of Russian Academy of Sciences, Malygina st., 86,
625026 Tyumen, Russia; pozhitkov-roma@yandex.ru (R.P.); fic@tmnsc.ru (A.Z.); ttrruubbaa@mail.ru (A.T.)
* Correspondence: moskovchenko1965@gmail.com; Tel.: +7-9224883211

**Abstract:** A study on the composition of snow allowed for a quantitative determination of pollutants deposited from the atmosphere. Concentrations of dissolved (<0.45 μm) and particulate fractions of 62 chemical elements were determined by ICP–MS and ICP–AES in 41 samples of snow from Tyumen (Russia). The background sites were characterized by a predominance of the dissolved phase of elements, except for Al, Sn, Cr, Co and Zr. The increased concentrations of dissolved Cd, Cu, Zn, Pb, Ni, As and Mo can be explained by a long-range atmospheric transport from the sources located in the Urals. The urban sites showed multiple increases in particulate depositions and a predominance of the particulate phase, with a high degree of enrichment in many heavy metals. Sources of trace elements were determined according to the enrichment factor (EF). Highly enriched elements (Pb, Sb, Cd, Ag, Mo, As, Zn and Cu) with an EF > 100 were emitted from anthropogenic sources. According to the potential ecological risk index (RI), the worst ecological conditions were identified in Tyumen's historical center, industrial zone and along roads with the heaviest traffic. The data obtained in the present study allowed us to identify the most polluted parts of the city, which are located in the center and along the roads with the most intensive traffic. This research could offer a reference for the atmospheric pollution prevention and control in Tyumen.

**Keywords:** Western Siberia; snow pollution; trace metals and metalloids; atmospheric depositions; solubility



## 1. Introduction

Currently industrial production is accompanied by the emission and spread of massive quantities of trace metals and metalloids (TMMs) [1]. An accurate and complete emission inventory for atmospheric trace metals is needed for both the modeler community and policymakers to assess the current levels of environmental contamination by these pollutants, major emission sources and source regions, and the contribution of the atmospheric pathway to the contamination of terrestrial and aquatic environments [2]. Some of Russia's large industrial regions are the sources of hazardous contamination of the environment by TMMs on a global scale. For example, TMMs emissions from the metallurgical plants of the Kola Peninsula and the Norilsk Industrial District, which have been described in several studies [3–6], have been shown to cause pollution of Arctic ecosystems [7,8]. However, there have been very few studies on the contamination processes in other regions of Russia, in particular, Western Siberia and the Urals.

Snowpack is often used in studies of atmospheric pollution. Insoluble aerosol particles, as well as soluble compounds, including various pollutants, are washed out of the atmosphere by snow, which acts as a reservoir of TMMs accumulation during the winter period [9–11]. When surfaces are covered with snow, the influence of soils on the formation of atmospheric dust aerosols is minimized, which can allow for accurate determinations of anthropogenic contributions and assessments of TMMs deposition rates in winter. Thus, the analysis of the chemical composition of snow provides useful information on aerosol

chemistry and long-range distribution patterns of anthropogenic substances emitted into the atmosphere [12,13].

Pollution by TMMs creates the greatest ecological hazard in highly populated cities. At the present time, 75% of the Russian population lives in cities [14]. Urbanization leads to the concentration of people, enterprises producing goods and services, and transport flows within cities, which consequently become the centers of environmental pollution. Most Russian cities, where ecological conditions have been assessed, are regarded as geochemical anomalies with highly polluted air, soils, water bodies and bottom sediments [15]. The majority of pollutants within such urban environments is deposited from the atmosphere [16]. Particles suspended in the air have high contents of TMMs which are deposited on the soil surface and cause deterioration of its properties. Atmospheric deposition is considered to be a major source of toxic TMMs, such as Cd, Cu Zn, Hg and Pb, for ecosystems [17,18]. Thus, an understanding of the impurities in natural snow is important in order to determine the air quality and to inform the pollution of the environment [19–21]. Since urban environments are heavily affected by anthropogenic pollutants, the quantitative analysis of TMMs in the snow cover can be of crucial importance [22].

In Siberia, snowpack is present for a period of 5–9 months. Such a long period of TMMs accumulation in snow allows for an accurate evaluation of the levels and sources of pollution. Snowpack is well known to be an informative object for evaluating aerogenic contaminants [23,24]. Unfortunately, the data on trace elements in snow meltwater collected from large and geographically homogeneous territories of Western Siberia are limited [11]. However, those limited data do demonstrate a significant impact of anthropogenic activities on the snow composition. In particular, it has been shown that there are latitudinal gradients of concentrations of certain TMMs within snowpack; for example, concentrations of Pb, Zn, Cd, V, Co and Sr decrease in a northward direction due to the location of the main industrial plants in the south [25]. It has also been suggested that the development of oil and gas fields in Western Siberia has caused TMMs accumulation in snow [26,27]. Anthropogenic impacts result in an enrichment of the particulate fraction of snow by many TMMs as compared to their levels in soils [11,28,29].

To date, the accumulation of major elements and TMMs present within the snowpack of cities of Western Siberia, as well as the interregional transmission of pollutants, has been poorly studied. Several studies of TMMs in the particulate fraction of snow have been conducted in Tomsk and Novosibirsk. It has been shown that Tomsk thermal power plants induce an enrichment of dust aerosols in Hg, Ba, Sb, La, As and Sr [30]. Moreover, in Novosibirsk, high TMMs concentrations in the particulate fraction of snow have been shown to have a technogenic origin and indicate a potentially higher toxicity of wintertime aerosols as compared to summertime aerosols [31].

Tyumen is one of the largest and rapidly developing cities of Western Siberia. The rapid development of Tyumen has been primarily due to the discoveries of numerous deposits of petroleum and natural gas in the north of Western Siberia during the second half of the 20th century. The population of Tyumen City was 150 thousand people in the early 1960s, which increased to around 500 thousand in the early 2000s and to over 800 thousand at the present time. In the late 1990s, it was revealed that the operation of the Tyumen thermal power station had caused the accumulation of Mn, Cr, Ni, Pb and V in the snowpack [32]. Since that time, the city's population has grown significantly and the distribution and specialization of industries have developed with the addition of new enterprises, including ferrous metallurgy and petroleum refining. An assessment of soil composition in Tyumen has shown increased concentrations of V, Cr, Co, Ni, Cu and Zn near to main roads and metalworking plants and anomalies of As and Pb near the operating facilities of electric storage battery plants [33]. The worsening of the environmental conditions in Tyumen necessitates research on the sources of pollutants and an ecological assessment, which can be carried out by analyzing the snowpack.

The present study had the following aims:

- To identify the basic distribution of the major and trace elements of snow,

- To evaluate the ratios of particulate and dissolved forms of TMMs,
- To assess the ecological risks associated with TMMs pollution of the atmosphere.

The originality of the present study is due to the combination of (i) the analysis of the snowpack composition within such a large Siberian industrial center as Tyumen and (ii) the assessment of both dissolved + particulate forms of major and trace elements in snow samples.

The proportions of dissolved and particulate forms of elements in snow meltwater allow for the determination of emission sources [15]. Elements associated with anthropogenic and marine sources are in general soluble upon thawing, elements from crustal sources are insoluble, elements with mixed sources tend to show intermediate behavior with average solubilities ranging from 27% to 96% for different elements [34]. In addition, the proportions of dissolved and particulate forms of elements in snow meltwater predetermine either their accumulation in soil or migration with meltwater into streams and rivers [35], which allows for the prognosis of contamination consequences.

## 2. Materials and Methods

### 2.1. Study Area and Sampling

Tyumen is located in the southwestern part of the West Siberian Plain, at the southern margin of the taiga zone (Figure 1).

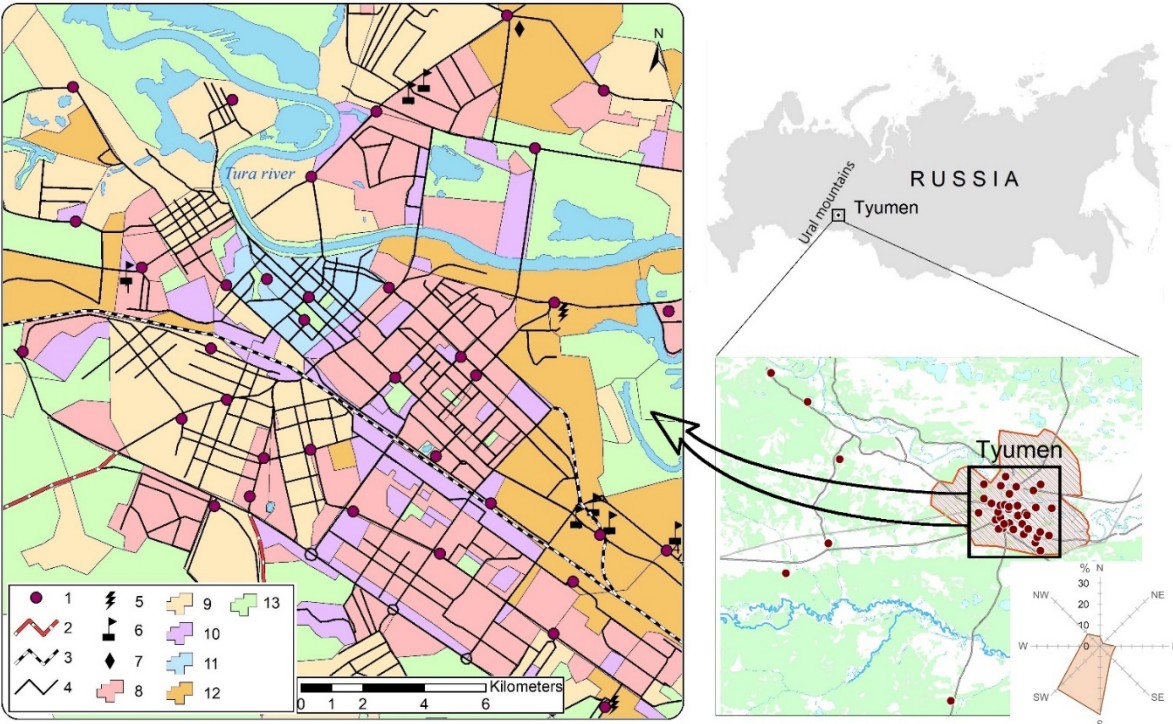

**Figure 1.** Sampling sites and land-use areas within the city of Tyumen, Russia: 1—sampling sites; 2—main federal roads; 3—Trans-Siberian railway; 4—main city roads; 5—power plants; 6—mechanical engineering and metalworking facilities; 7—oil refining and plastic production facilities; 8—high-rise residential area; 9—low-rise residential area; 10—modern business zones; 11—historical center; 12—industrial zones; 13—recreational and unbuilt zones. The diagram in the lower right corner shows the wind directions within the period from November 2019 to February 2020.

Tyumen has a cold humid continental climate, with mean monthly temperatures ranging from −22 °C in January to +17 °C in July and a mean annual precipitation of 480 mm. On average, stable snow cover is present for five months of the year, from early November to early April. The cold period is characterized by a predominance of southern and western winds, which carry pollutants to the north and northeast [36,37]. The winter season of 2019–2020 was not exceptional in this respect, as it was dominated by southern,

southwestern and south–southwestern winds. A flat surface topography within the city minimizes the influence of the orographic factor on the translocation of pollutants. Winds of different directions provide for the redistribution of pollutants within the city and beyond [38].

Tyumen is a large center of transportation and trade with a well-developed building industry. Tyumen now has a number of industrial plants producing machinery, equipment and building materials, petroleum refineries and power stations. Moreover, the city currently has 385 thousand motorized vehicles in use and a network of roads totaling 1241 km in length [39]. Major transport routes pass through Tyumen to northern regions of Western Siberia, where the largest oil and gas fields are located. Exhaust gases from motorized vehicles make up more than 80% of the total emissions of pollutants into the atmosphere within the city [40]. Traffic on the busiest roads reaches 8000 vehicles per hour [41]. Most (89.4%) of the motorized vehicles have gasoline engines. The geochemical condition of the city is estimated as hazardous and ecologically unfavorable residential areas have been identified [39–41].

### 2.2. Sample Collection

Snow samples were taken within a period from the 17th to 20th of February 2020, when the air temperature in Tyumen was close to $0°$ C, which was significantly above its mean range of $-15$ to $-10\ °C$. On average, the snow cover of the city disappears on the 9th of April [37]. However, the winter season of 2019–2020 was abnormally warm, with short-term warmings throughout January and February. Therefore, sampling was moved to the earlier dates in order to take samples of snowpack prior to the beginning of its thawing. A total precipitation of 106 mm was recorded over the period from November 2019 to February 2020 [36], which is higher than the mean of 91 mm over the period 1960–2019 [37].

Snow was sampled from within the Tyumen City and from background sites at distances of 20–35 km to the west and southwest from the city (see Figure 1). Taking into account that winds from the south prevailed during the study period, the locations of background sampling sites excluded any contamination of snow by pollutants from Tyumen.

During sampling, the land-use type of each surveyed area was taken into account. The functional zones (different land-use areas) within Tyumen are not clearly separated, but form a mosaic pattern of industrial, residential, commercial and recreational areas within the city. However, different types of land use were recorded at the sampling sites as follows:

(1) The historical center with buildings that have existed since the 17th century and are now used by social and administrative organizations;

(2) Low-rise residential area that have existed since the 19th century;

(3) High-rise residential area that have been constructed within a period from the 1950s to the present time and that now house the major part of the Tyumen population;

(4) Modern business zones;

(5) Industrial zones;

(6) Transport zones affected by road traffic (located between main roads and buildings of various usage). Sampling within transport zones was carried out at distances of more than 15 m from roads in order to exclude direct contamination by road dust. The sampled snowpack was visibly undisturbed by any human activity.

Samples of snow were taken by using a VS-43 snow gauge, which is a widely used instrument for meteorological observations in Russia. This snow gauge consists of an aluminum pipe (sampling cylinder) with the cross-section area of 50 cm$^2$ and a measurement scale for determining the depth of snow. In order to exclude contamination of samples by soil particles, we removed the lowest 3 cm of snow that was directly adjacent to the ground. Samples were placed in 24 dm$^3$ plastic containers with lids, which were washed with distilled water prior to sampling. Closed containers with samples were transported to the environmental chemistry laboratory. In total, 35 samples were taken from the city of Tyumen, and 6 samples were from background sites. Locations of sampling sites are shown

in Figure 1. A detailed description of the sampling sites is presented in Supplementary Materials Table S1.

### 2.3. Analytical Procedures

The snowpack samples were melted at room temperature. In the melted samples, pH values were measured by using HI83141 and HM-500 HydroMaster ionometers, and salinity was determined by using a COM-100 conductometer. Afterwards, the partitioning of elements between the dissolved phase and the solid phase was performed in order to separately analyze TMMs in water-soluble mobile forms and forms bound with mineral and organomineral compounds [42]. The partitioning was achieved by passing the melted samples through pre-weighed Millipore ash-less nitrocellulose filters (0.45 μm pore size). Around 1.5–2 liters of meltwater was filtered to obtain a sufficient amount of solid residue from each sample. The filters were dried at 95 °C and weighed again for determination of the mass of trapped solids.

Contents of trace elements (Li, Be, Sc, V, Cr, Ti, Mn, Co, Ni, Cu, Zn, Ga, As, Se, Rb, Sr, Y, Zr, Nb, Mo, Rh, Pd, Ag, Cd, Sn, Sb, Te, Cs, Ba, La, Ce, Pr, Nd, Sm, Eu, Gd, Tb, Dy, Ho, Er, Tm, Yb, Lu, Hf, Ta, W, Re, Ir, Hg, Tl, Pb, Bi, Th and U) and major elements Na, Mg, Al, P, S, K, Ca and Fe (in weight percent oxide for the particulate fraction) were determined by inductively coupled plasma–mass spectrometry ICP–MS (Thermo Elemental-X7 spectrometer, USA) and inductively coupled plasma–atomic emission spectrometry ICP–AES (Thermo Scientific iCAP-6500 spectrometer, USA).

The solids trapped on filters were decomposed by an acid digestion in open system. The samples were placed in Teflon beakers (volume 50 mL), and 0.1 mL of a solution containing 8 μg $L^{-1}$ 145Nd, 61Dy and 174Yb was added and moistened with several drops of deionized water. Then 0.5 mL of $HClO_4$ (perchloric acid fuming 70% Supratur, Merck), 3 mL HF (Hydrofluoric acid 40% GR, ISO, Merck), 0.5 mL of $HNO_3$ (nitric acid 65%, maximum 0.0000005%% GR, ISO, Merck) and evaporated until intense white vapors appeared. The beakers were cooled, their walls were washed with water, and the solution was again evaporated to wet salts. Then 2 mL of HCl (hydrochloric acid fuming 37% OR, ISO, Merck) and 0.2 mL of 0.1M $H_3BO_3$ solution (analytical grade) were added and evaporated to a volume of 0.5–0.7 mL. The resulting solutions were transferred into polyethylene bottles, 0.1 mL of a solution containing 10 mg $L^{-1}$ In (internal standard) was added, diluted with deionized water to 20 mL, and analysis was performed. As control samples in Teflon beakers, the above procedures were performed without samples, and the resulting solutions were used as controls.

The accuracy of the analytical procedure was confirmed by analysis of Standard reference material Gabbro Essexit STD-2A (GSO 8670-2005). The methods, recoveries and analytical results of certified reference material for the solid phase are given in Supplementary Materials Table S2.

Element concentrations in filtrates were determined by a quantitative method, using reference solutions with 0.5, 1 and 10 μg $L^{-1}$ concentrations of studied elements. In the studied samples, element concentrations were calculated by the spectrometer software. The accuracy of determinations was verified by using the standard sample of "Trace Metals in Drinking Water" produced by High-Purity Standards (Charleston, VA, USA). In addition, the accuracy of the filtrate analyses was verified by comparing the results of ICP–MS and ICP–AES determinations of Li, Al, Mn, Cu, Zn, Sr and Ba. In all cases, differences between the values determined by the two methods were within the standard errors of those methods. The detection limit (DL) was calculated by using the following equation:

$$DL = Ci + 3SD$$

where Ci is the mean value of element concentration in deionized water s and SD is the standard deviation for repeated measurements. The methods, recoveries and analytical results of certified reference material for the dissolved phase were given in Supplementary Materials Table S3.

### 2.4. Calculations and Data Analysis

Calculations of statistical parameters (arithmetic and geometric mean, standard deviation) were performed by using Microsoft Excel. Total concentrations of elements were calculated with the use of geometric mean values, because the compilation of Normal Probability Plots showed an absence of normal distribution for most of the analyzed elements. Geometric means have been previously applied in the assessment of regional variations in the snowpack composition within Western Siberia [11]; therefore, geometric mean values in the present study were the most appropriate for the comparability of the data obtained. In order to evaluate the intensity of pollution, we calculated the contamination factor (CF) and the enrichment factor (EF) [43] with the help of the following formulas:

$$CF = Cn/Cb$$

where Cn is the measured concentration of the element in the sample and Cb is the background value of the element.

$$EF = (Cn/CAl)sample/(Cn/CAl)crust$$

where (Cn)sample is the measured concentration of the element of interest, (Cn) crust is the concentration of the same element in the Earth's crust, and CAl is the concentration of the reference element in the same sample and the Earth's crust. Unfortunately, available data on soil composition within the Tyumen area include only very few trace elements, which makes it impossible for us to compare the enrichment of particulate matter of snow with that of local soils. Determination of regional baseline concentrations of TMMs in soils still remains an unresolved problem due to the vastness of the territory of Western Siberia, highly variable TMMs concentrations in soils and only small datasets analyzed to date [44].

For the EF calculations, we used data on average element concentrations in the upper part of the continental Earth's crust according to Reference [45].

The CF and EF indices have different applications. The CF values were calculated with the use of the local background values and allow for the evaluation of the influence of urban pollution sources. The EF calculations were based on comparisons with mean values in the Earth's crust which allowed us to evaluate the pollution level at a global scale. Specifically, following the common practice in this field, the normalization to the upper Earth's crust allowed us to assess the enrichment of the atmospheric aerosol and estimate the long-range atmospheric transport of soluble and particulate forms of elements. According to References [46,47], element sources are classified into three groups depending on EF values as follows: EF < 10 is considered to be of a crustal origin, without enrichment; 10 < EF < 100 has a mixed (crustal and anthropogenic) origin; EF > 100 indicates man-made air pollution.

Concentrations in the solid phase were recalculated in relation to the volume of filtered meltwater to allow for a direct comparison with the solution data. Then the percentage of soluble elements (PSE) was calculated as follows:

$$PSE = (Cn\ dissolved)/(Cn\ total)\cdot100\%$$

Geometric means were used in PSE calculations. The potential ecological risk index $E_r^i$, which characterizes a degree of the ecological risk of a single element [48], was calculated by using the following equation:

$$E_r^i = CF \cdot T_r^i$$

where CF is the contamination factor and $T_r^i$ is the toxicity response coefficient.

In this study, we used the response $T_r^i$ values according to References [48,49], as follows: Zn, Mn, Fe, W, Sr = 1; Cr, Mo, Sn, Sb = 2 ; Pb, Cu, Co, Ni = 5; As = 10, Cd = 30. For risk assessments we adopted the following gradation: $E_r^i$i < 40 describes low risk; 40 < $E_r^i$ < 80 indicates moderate risk; 80 < $E_r^i$ < 160 indicates considerable risk; 160 < $E_r^i$ < 320 indicates high risk; $E_r^i$ > 320 indicates extreme risk [49,50].

The total potential ecological risk index (*RI*) characterizes the overall degree of the ecological risk of all metals under investigation [48].

$$RI = \sum E_r^i$$

where $E_r^i$i is a potential ecological risk index of a single element. Risk levels were graded as follows: RI < 150, low; 150 < RI < 300, moderate; 300 < RI < 600, considerable; RI > 600, high ecological risk.

In order to assess the risks associated with different forms of elements, we calculated $E_r^i$ and *RI* values separately for dissolved and particulate forms and their total concentrations.

## 3. Results and Discussion

### 3.1. Salinity, pH, and Dust Content

Meltwaters from the background sites had an acid reaction, which is typical for snowpack within the taiga zone. Their mean pH value (4.7) was lower than the average meltwater pH (5.3) within the Khanty-Mansi Autonomous Okrug [26]. The acidification of snow around Tyumen can be explained by long-range atmospheric transport from the Urals' metallurgical plants. It has been shown that sulfur compounds emitted from smelters and also from oil-producing enterprises, where they are synthesized as a by-product of gas combustion, are the major factors responsible for water acidification in Russia [51]. Considering that Tyumen is affected by predominantly southern and southwestern winds during the winter (Figure 1), the acidifying agents were likely to be transported from the Southern Urals, where numerous metallurgical districts (Chelyabinsk, Magnitogorsk, Sibay, etc.) are located at distances of 300–500 km to the southeast of Tyumen. Massive emissions of sulfur dioxide from metallurgical plants of the Urals have been shown to have the potential to cause acidification of atmospheric precipitation [52]. For example, the air within the Karabash copper smelter area is characterized by an $SO_2$ concentration of 20,000 $\mu g\ m^{-3}$, which is far higher than the standard $SO_2$ concentration of 500 $\mu g\ m^{-3}$ according to the WHO air quality guidelines [53]. The development of a weakly acid reaction of snow has been reported in the eastern part of the Sverdlovsk (Yekaterinburg) Oblast, which borders the Tyumen Oblast from the west [52]. The long-range atmospheric transport of TMMs-enriched aerosols from the Urals' metallurgical plants has also been shown to affect the area of Arctic seas [54].

The mean meltwater salinity of 9.5 $mg\ L^{-1}$ was typical for unpolluted areas of Western Siberia. A dissolved salt content of less than 15 $mg\ L^{-1}$ of atmospheric precipitation can be regarded as a regional background in Russia [55].

The content of particulate matter in meltwaters from the background sites varied from 4 to 10 $mg\ L^{-1}$, which is slightly higher than the values reported for the Arctic snow cover [29,56]. It is probable that the dust in the studied background samples is sourced mainly from soils of the steppe areas that are located south of Tyumen, which have a relatively thin snow cover [11]. Deflation of soil particles under such conditions can be quite intensive even during the winter [57]. It is also not excluded that some aerosols can be carried from the industrial regions of the Middle and Southern Urals.

The composition of snow meltwaters within the city of Tyumen was significantly different from that within the background area. There was a significant increase in the mean pH value (6.3) of the city snow as compared to the mean background value (4.7), although the reaction of most of the samples from the city remained weakly acid with only a few samples with neutral and weakly alkaline reactions. The observed alkalinization of the city snow was due to the deposition of dust from building materials, which predominantly consist of carbonates [57].

Snow meltwaters in the city had a mean salinity of 68.1 $mg\ L^{-1}$, which was more than 7 times higher than that in the background. The maximal content of dissolved salts (202–564 $mg\ L^{-1}$) was observed near the roads treated with de-icing agents, which are mainly composed of technical salt NaCl. For comparison, the salinity of snow meltwater

within Moscow is 4 times as high as the mean background value (23 mg L$^{-1}$) [58]. The content of particulate matter in meltwaters from the city varied from 9 to 121 mg L$^{-1}$, with a mean value of 37 mg L$^{-1}$, which was 5 times as high as the background value.

### 3.2. Elemental Composition

The elemental composition of snow meltwater from the background sites is presented in Table 1. In the dissolved phase, Ca was predominant and other major elements were arranged in the following decreasing order of concentrations: Na > S > Mg > K > Fe > Al. Interestingly, major elements in the global river discharge form a nearly identical order based on the values of the dissolved transport index (DTI) as follows: Na ≈ Ca > Mg > K > P > Fe > Al [59]. The DTI is a ratio of dissolved forms of elements to their total content, which in the global river discharge depends on the solubility of those elements. The similarity of the concentration orders led us to conclude that the element contents in the snowpack from the study area were predetermined by their solubility, with only insignificant influences of anthropogenic and other factors (e.g., a predominance of either continental or oceanic aerosols).

**Table 1.** The elemental composition (µg L$^{-1}$) of meltwater from the background sites.

| Elements | Dissolved | | | | Particulate | | | | Total | PSE, % |
|---|---|---|---|---|---|---|---|---|---|---|
| | DL | Mean | SD | Geometric Mean | DL | Mean | SD | Geometric Mean | | |
| Li | 0.007 | 0.14 | 0.095 | 0.11 | 0.0014 | 0.005 | 0.0029 | 0.005 | 0.12 | 95.7 |
| Na | 5 | 402 | 374 | 282 | 1.3 | 2.8 | 0.6 | 2.7 | 284.7 | 98.9 |
| Mg | 4 | 131 | 64 | 119 | 1.5 | 12.4 | 12.5 | 9.5 | 128.5 | 92.4 |
| Al | 1 | 10.2 | 4.9 | 8.9 | 0.9 | 11.7 | 5.4 | 10.9 | 19.8 | 45.2 |
| P | 16 | | nd | | 0.25 | 0.76 | 0.64 | 0.63 | 0.63 | nd |
| S | 15 | 355 | 68 | 350 | 1.5 | 7.2 | 0.5 | 7.1 | 357.1 | 98.0 |
| K | 5 | 72.3 | 18.7 | 70.3 | 0.9 | 4.1 | 1.1 | 3.9 | 74.2 | 94.7 |
| Ca | 6 | 595 | 135 | 581 | 2.5 | 5.6 | 0.3 | 5.6 | 586.6 | 99.0 |
| Ti | 0.8 | | nd | | 0.036 | 0.081 | 0.075 | 0.065 | 0.065 | nd |
| V | 0.07 | 0.152 | 0.088 | 0.11 | 0.025 | 0.086 | 0.055 | 0.072 | 0.18 | 59.4 |
| Cr | 0.6 | | nd | | 0.05 | 0.27 | 0.130 | 0.25 | 0.25 | nd |
| Mn | 0.07 | 7.3 | 6.17 | 5.89 | 0.05 | 0.15 | 0.073 | 0.14 | 6.04 | 97.6 |
| Fe | 4 | 20.2 | 7.53 | 18.9 | 3.0 | 14.0 | 9.48 | 12.2 | 31.1 | 60.8 |
| Co | 0.1 | | nd | | 0.003 | 0.009 | 0.006 | 0.008 | 0.008 | nd |
| Ni | 0.3 | 0.95 | 0.46 | 0.80 | 0.032 | 0.161 | 0.099 | 0.15 | 0.95 | 84.2 |
| Cu | 0.3 | 4.68 | 2.53 | 4.09 | 0.032 | 0.091 | 0.046 | 0.082 | 4.17 | 98.1 |
| Zn | 0.7 | 13.2 | 5.51 | 12.2 | 0.14 | 0.535 | 0.083 | 0.530 | 12.7 | 96.1 |
| As | 0.06 | 0.505 | 0.13 | 0.49 | 0.031 | 0.046 | 0.015 | 0.045 | 0.54 | 90.7 |
| Rb | 0.006 | 0.100 | 0.023 | 0.098 | 0.0023 | 0.009 | 0.005 | 0.008 | 0.11 | 89.1 |
| Sr | 0.08 | 1.505 | 0.37 | 1.46 | 0.014 | 0.044 | 0.010 | 0.043 | 1.51 | 96.7 |
| Y | 0.002 | 0.004 | 0.002 | 0.003 | 0.0005 | 0.004 | 0.002 | 0.004 | 0.007 | 42.9 |
| Zr | 0.004 | | nd | | 0.0005 | 0.032 | 0.012 | 0.030 | 0.030 | nd |
| Nb | 0.004 | | nd | | 0.0005 | 0.0044 | 0.0013 | 0.0042 | 0.0042 | nd |
| Mo | 0.008 | 0.051 | 0.016 | 0.049 | 0.0002 | 0.0094 | 0.0025 | 0.0092 | 0.058 | 84.5 |
| Ag | 0.004 | 0.022 | 0.016 | 0.017 | 0.0002 | 0.010 | 0.008 | 0.008 | 0.025 | 68.0 |
| Cd | 0.004 | 0.096 | 0.030 | 0.092 | 0.0004 | 0.001 | 0.002 | 0.001 | 0.093 | 99.0 |
| Sn | 0.008 | 0.076 | 0.043 | 0.066 | 0.0034 | 0.137 | 0.149 | 0.091 | 0.157 | 42.0 |
| Sb | 0.004 | 0.104 | 0.029 | 0.100 | 0.0015 | 0.003 | 0.001 | 0.003 | 0.104 | 96.2 |
| Cs | 7·10$^{-4}$ | 0.0058 | 0.0015 | 0.0057 | 0.0004 | 0.0009 | 0.0004 | 0.0009 | 0.0065 | 87.7 |

**Table 1.** *Cont.*

| Elements | Dissolved | | | | Particulate | | | | Total | PSE, % |
|---|---|---|---|---|---|---|---|---|---|---|
| | DL | Mean | SD | Geometric Mean | DL | Mean | SD | Geometric Mean | | |
| Ba | 0.05 | 3.20 | 1.15 | 4.69 | 0.005 | 0.08 | 0.04 | 0.08 | 4.77 | 98.3 |
| La | 0.003 | 0.0100 | 0.0050 | 0.0090 | 0.0003 | 0.0069 | 0.0027 | 0.0065 | 0.016 | 58.1 |
| Ce | 0.002 | 0.014 | 0.009 | 0.012 | 0.0003 | 0.013 | 0.005 | 0.012 | 0.024 | 50.0 |
| Pr | $4 \cdot 10^{-4}$ | 0.0017 | 0.0010 | 0.0015 | 0.0003 | 0.0010 | 0.0005 | 0.0009 | 0.0024 | 62.5 |
| Nd | 0.002 | 0.0072 | 0.0026 | 0.0068 | 0.0003 | 0.0058 | 0.0037 | 0.0049 | 0.0118 | 57.6 |
| Sm | $9 \cdot 10^{-4}$ | 0.0044 | 0.0042 | 0.0032 | 0.0005 | 0.0008 | 0.0005 | 0.0008 | 0.0039 | 82.1 |
| W | 0.003 | 0.012 | 0.007 | 0.011 | 0.0005 | 0.003 | 0.002 | 0.002 | 0.012 | 91.7 |
| Pb | 0.02 | 1.34 | 0.76 | 1.12 | 0.002 | 0.13 | 0.10 | 0.10 | 1.23 | 91.1 |
| Bi | 0.001 | 0.011 | 0.008 | 0.008 | 0.0003 | 0.0010 | 0.0004 | 0.0010 | 0.0090 | 88.9 |
| Th | $9 \cdot 10^{-4}$ | 0.0019 | 0.0006 | 0.0019 | 0.0002 | 0.0011 | 0.0006 | 0.0010 | 0.0028 | 67.9 |
| U | $5 \cdot 10^{-4}$ | 0.0015 | 0.0005 | 0.0014 | 0.0002 | 0.0005 | 0.0003 | 0.0004 | 0.0019 | 73.7 |

Note: concentrations of Be, Sc, Ga, Se, Rh, Pd, Te, Eu, Gd, Hg, Tb, Dy, Ho, Er, Yb, Hf, Ta and Lu were below their detection limits; nd—not detected in more than 80% of samples; total is the sum of the geometric mean.

The majority of elements within the background area occurred mainly in their dissolved forms, the proportions of which varied from 43% (Sn) to 98.9% (Na) (see Table 1). Al and Sn were represented mostly by particulate forms. Aluminum belongs to very weakly soluble elements and Sn is similar to Fe and Al in terms of solubility [60]. Concentrations of Ti, Cr, Co and Zr in the dissolved phase were below their detection limits in more than 80% of samples, which is also indicative of the predominance of their particulate forms. Approximately equal proportions of dissolved and particulate forms were found in Fe, V and some of the rare-earth elements (Ce, Nd and La). A majority of TMMs (Cu, Zn, Ni, As, Mo, Cd, Pb, Sr, Ba and Bi) were found predominantly in the dissolved phase (83–98% of their total concentrations).

According to Reference [11], Na, Ca, Sr and Cd in snow meltwaters from West-Siberian areas to the north of Tyumen are found mostly in dissolved forms; K, As, Zn and Ba are equally divided between the dissolved and particulate phases; and other elements occur predominantly in the particulate phase. The background sites near Tyumen were distinguished by much higher proportions of dissolved forms of elements. We attribute such a discrepancy to the fact that the study [11] included not only background, but also areas affected by local pollution sources. Even low-intensity human impact can result in an increased deposition of dust aerosols and shifts the balance of dissolved and particulate phases.

Studies conducted in different regions have demonstrated that the proportions of dissolved and particulate phases are predetermined by their emission sources. According to Reference [61], Al, Ca, Fe, K, Mg and Na originate from natural sources, whereas trace metals, including Cd, Cu, Ni, Pb and Zn, are primarily associated with anthropogenic activities. An analysis of the literature showed that Al, Fe, Cr and Co generally prevail within conventional background areas located at significant distances from any sources of industrial emissions. For example, atmospheric aerosols from the Northern Atlantic region have been shown to be dominated by the particulate fractions of Al, Fe and Co [62]. A study on snow composition in China showed that the least soluble elements were Fe, Al and Cr with a solubility of less than 30% [46]. Likewise, in Quebec (Canada), a study has shown that Fe, Ni, Cr, Co, V, La, Be, Ce, Tl, Y and Rb are found predominantly in the particulate fraction [10]. In that study the assemblage of elements found mainly in the particulate fraction is very similar to that in the present study, except for Ni, which near Tyumen mostly occurred in the dissolved form, which substantiates the conclusion about its anthropogenic origin.

Assemblages of elements that occur predominantly in dissolved forms in different background areas of the world can vary depending on their anthropogenic emission

sources, the composition of local soils and the direction of atmospheric transport. In China, the most soluble elements (with solubilities reaching 60–70 %) are As, Mn, Cu, Zn Cd and Se emitted from regional coal-fired power plants and municipal solid waste incineration [46]. In Canada, the influence of a metal smelter resulted in the predominance of soluble Cu, Zn, Cd, Pb, S, Sr and Sb [10].

It can be stated that background areas, where dust deposition is insignificant, are characterized by the predominance of dissolved fractions of ecologically hazardous TMMs (Cd, Pb, Ni, Cu, Zn, As and Mo), except for Fe, Co and Cr that occur mainly in particulate forms. The low solubility of Fe, Co, Cr and Al is explained by their strong bonds with the crystalline lattice of aluminosilicate minerals. The elements such as Zn, Cu, Mn, Ni and V bound with carbonate minerals are highly soluble [63]. Therefore, the quantities and ratios of dissolved and particulate fractions of trace elements are predetermined by the mineralogical composition of dust aerosols.

In order to assess the local geochemical background of Tyumen, we compared the data obtained from the background sites with average values for Western Siberia [11]. Most of major and trace elements around Tyumen have higher concentrations in the dissolved phase as compared to the regional average (Figure 2). The most significant differences were observed in the elements connected with carbonate dust aerosols of natural origin (Mn and V) and the elements emitted into the atmosphere from industrial plants and transportation (Pb, Cd, Cu, Sb, Ni and As). It should be noted that the metallurgical districts of the Southern Urals are characterized by a similar assemblage of pollutants. The particulate deposition rates and concentrations of Cu, Zn, Pb, Cd, As, Se, Bi, Sb and Sn within the industrial mining districts of the Southern Urals exceed those within conventional background areas by three orders of magnitude [64]. In Chelyabinsk (Southern Urals), Cd, Cu, Mn, Ni, Pb, Sr and Zn enrichment of atmospheric dust is caused by the metallurgical industry [65]. Therefore, high concentrations of dissolved Pb, Cd, As, Cu, Sb and Ni, as the snow acidification within the study area, can be associated with the influence of the industrial regions located in the middle and Southern Urals, taking into account the predominance of southern and southwestern winds within the study area (see Figure 1). Previously, it has been shown that some of the pollutants present in the south of the Tyumen Oblast have been carried from the Urals' large industrial plants [66]. Thus, the concentration of dissolved forms of elements is an effective indicator of the long-range transport of aerosols.

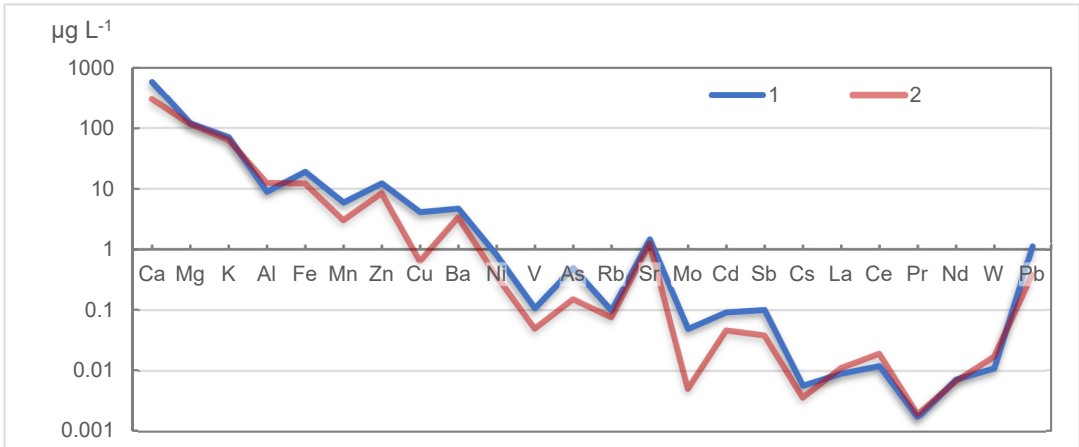

**Figure 2.** Concentrations of elements in the dissolved phase: 1—background sites near Tyumen (our data); 2—mean regional concentrations within Western Siberia [8].

The elemental composition of snow meltwater within Tyumen, with the partitioning between dissolved and particulate fractions, is presented in Table 2.

**Table 2.** The elemental composition ($\mu g\ L^{-1}$) of meltwater within the city of Tyumen.

| Elements | Dissolved | | | Particulate | | | Total | PSE, % |
|---|---|---|---|---|---|---|---|---|
| | Mean | SD | Geometric Mean | Mean | SD | Geometric Mean | | |
| Li | 0.88 | 2.73 | 0.31 | 0.78 | 0.95 | 0.27 | 0.58 | 52.5 |
| Be | | nd | | 0.047 | 0.054 | 0.018 | 0.018 | nd |
| Na | 15612 | 24849 | 9115 | 375 | 458 | 136 | 9251 | 98.5 |
| Mg | 1473 | 867 | 1226 | 11608 | 15962 | 3219 | 4445 | 27.6 |
| Al | 14 | 9 | 11 | 2472 | 2937 | 836 | 847 | 1.3 |
| P | 19.7 | 19.2 | 14.5 | 82.4 | 156.6 | 25.7 | 40.2 | 36.1 |
| S | 1123 | 396 | 1070 | 187 | 231 | 80 | 1150 | 93.0 |
| K | 393 | 350 | 293 | 345 | 403 | 138 | 431 | 67.9 |
| Ca | 6674 | 4105 | 5753 | 2953 | 4348 | 777 | 6530 | 88.1 |
| Sc | | nd | | 1.11 | 1.44 | 0.33 | 0.33 | nd |
| Ti | | nd | | 184.7 | 260.6 | 54.0 | 54.0 | nd |
| V | 0.16 | 0.07 | 0.14 | 6.71 | 8.09 | 2.16 | 2.30 | 6.1 |
| Cr | | nd | | 81.2 | 109.8 | 24.7 | 24.7 | nd |
| Mn | 1.8 | 1.4 | 1.4 | 98.9 | 123.4 | 31.5 | 32.9 | 4.3 |
| Fe | 10.9 | 9.3 | 6.2 | 5728 | 7882 | 1638 | 1644 | 0.4 |
| Co | | nd | | 5.71 | 8.03 | 1.58 | 1.58 | nd |
| Ni | 4.3 | 2.5 | 3.7 | 93.4 | 129.9 | 27.8 | 31.5 | 11.7 |
| Cu | 4.9 | 3.6 | 3.6 | 24.2 | 37.1 | 8.0 | 11.6 | 31.3 |
| Zn | 10.0 | 4.0 | 9.1 | 59.1 | 70.6 | 25.0 | 34.1 | 26.7 |
| Ga | | nd | | 0.46 | 0.56 | 0.15 | 0.15 | nd |
| As | 0.44 | 0.18 | 0.36 | 1.04 | 1.15 | 0.47 | 0.83 | 43.4 |
| Rb | 0.37 | 0.30 | 0.30 | 1.13 | 1.34 | 0.42 | 0.72 | 41.1 |
| Sr | 30.0 | 74.5 | 13.8 | 6.95 | 9.08 | 2.5 | 16.3 | 85.2 |
| Y | | nd | | 0.62 | 0.79 | 0.21 | 0.21 | nd |
| Zr | | nd | | 4.36 | 5.66 | 1.52 | 1.52 | nd |
| Nb | | nd | | 0.39 | 0.52 | 0.14 | 0.14 | nd |
| Mo | 0.24 | 0.09 | 0.22 | 0.47 | 0.75 | 0.16 | 0.38 | 57.9 |
| Ag | 0.026 | 0.014 | 0.021 | 0.035 | 0.036 | 0.018 | 0.04 | 52.5 |
| Cd | 0.050 | 0.019 | 0.045 | 0.097 | 0.11 | 0.038 | 0.083 | 56.3 |
| Sn | 0.079 | 0.063 | 0.054 | 1.46 | 2.21 | 0.55 | 0.61 | 8.9 |
| Sb | 0.35 | 0.20 | 0.28 | 1.28 | 1.69 | 0.46 | 0.74 | 38.4 |
| Cs | 0.022 | 0.032 | 0.015 | 0.080 | 0.095 | 0.030 | 0.044 | 34.1 |
| Ba | 59.3 | 96.6 | 35.9 | 24.3 | 31.4 | 8.2 | 44.1 | 81.4 |
| La | 0.010 | 0.006 | 0.009 | 0.73 | 0.89 | 0.27 | 0.28 | 3.2 |
| Ce | 0.014 | 0.010 | 0.011 | 1.61 | 2.0 | 0.57 | 0.585 | 1.9 |
| Pr | 0.0015 | 0.0008 | 0.0013 | 0.15 | 0.18 | 0.052 | 0.053 | 2.5 |
| Nd | 0.0059 | 0.0033 | 0.0051 | 0.543 | 0.65 | 0.200 | 0.205 | 2.5 |
| Sm | 0.0026 | 0.0015 | 0.0022 | 0.11 | 0.135 | 0.040 | 0.043 | 5.1 |
| Gd | 0.0015 | 0.0004 | 0.0015 | 0.096 | 0.12 | 0.034 | 0.036 | 4.2 |
| Er | 0.0015 | 0.00024 | 0.00075 | 0.063 | 0.077 | 0.022 | 0.022 | 3.4 |
| W | 0.11 | 0.11 | 0.09 | 1.34 | 1.81 | 0.36 | 0.44 | 20.5 |
| Pb | 0.34 | 0.30 | 0.21 | 13.1 | 20.8 | 4.6 | 4.80 | 4.4 |
| Bi | 0.0081 | 0.0071 | 0.0059 | 0.11 | 0.16 | 0.040 | 0.046 | 12.8 |
| Th | 0.0014 | 0.0004 | 0.0014 | 0.13 | 0.15 | 0.050 | 0.052 | 2.7 |
| U | 0.0043 | 0.0091 | 0.0024 | 0.084 | 0.105 | 0.028 | 0.030 | 8.0 |

Note: concentrations of Se, Rh, Pd, Te, Eu, Hg, Tb, Dy, Ho, Tm, Yb, Hf, Re, Ir, Ta Tl and Lu were below their detection limits; nd—not detected in more than 80% of samples; total is the sum of the geometric mean.

Concentrations of major elements decreased in the following order: Na > Ca > Mg > S > K > P > Al > Fe. More than 50% of Na, S, Ca and K were found in dissolved forms, as those elements are known to be the most active water migrants [60]. Hence, major element concentrations in the snowpack samples from both the city and background areas depend primarily on the element solubility. However, it should be noted that some samples had very high concentrations (up to 144 mg $L^{-1}$) of Na, which were indicative of the use of

de-icing agents, i.e., solid technical salt NaCl being the main agent applied to Tyumen's roads and pavements in order to remove snow and ice.

A prevalence of dissolved forms was observed in highly soluble trace elements (Mo, Cd, Li and Sr), as well as in Ba, which has a high solubility when the $SO_4^{2-}$ concentration is low [60]. Other TMMs were characterized by the predominance of particulate fractions; for example, Al, Fe, Ni, Cr, Co, Ti, Pb, Sn and Bi had less than 13% of dissolved forms (Figure 3). Similarly, urban snow of Moscow contains Sn, Ti, Bi, Al, W, Fe, Pb, V, Cr, Rb, Mo, Mn, As, Co, Cu, Sb and Mg mainly in particulate form, and Ca and Na in dissolved form [67]. Due to the increased rate of dust aerosol deposition within Tyumen as compared to that in the background area, the balance between dissolved and particulate forms of elements is shifted towards the predominance of particulates (Figure 3).

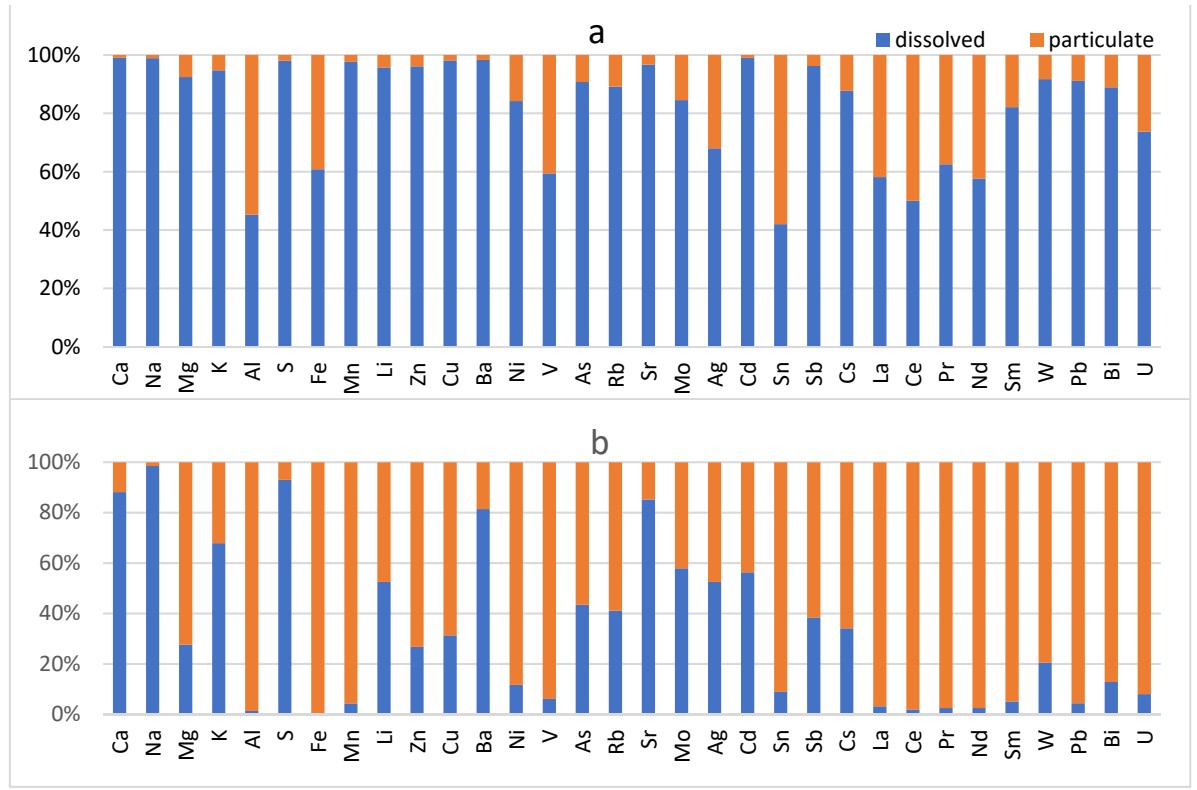

**Figure 3.** Proportions of dissolved (1) and particulate (2) forms of elements in the background area (**a**) and Tyumen City (**b**).

The values of CF, characterizing the surplus of element concentration within the city as compared to the background, were significantly higher in the particulate phase as compared to the dissolved phase (Figure 4). This is connected with a higher rate of dust deposition within the city, as well as a higher total concentration of TMMs in the dust. It should be mentioned that concentrations of dissolved forms of some elements (Mn, Fe, Co, Cu, Zn, As, Cd, Sn, Pb and Bi) within the city were lower than those in the background, with CF < 1 (Figure 4). This finding is paradoxical, taking into account that the deposition rate of metals in particulate forms had multiplied by tens and hundreds of times within the city. Nevertheless, similar situations have been reported in previous studies. For example, Viklander [68] detected a decrease in the dissolved Pb concentration in the sites affected by heavy traffic in comparison with that in background sites. Possible explanations can include either the coagulation of small (d = 0.8–1.5 μm) particles of dissolved fractions into larger (d = 50–600 μm) insoluble particles [69], or the adhesion of smaller to larger particles that would also decrease the content of dissolved forms [68].

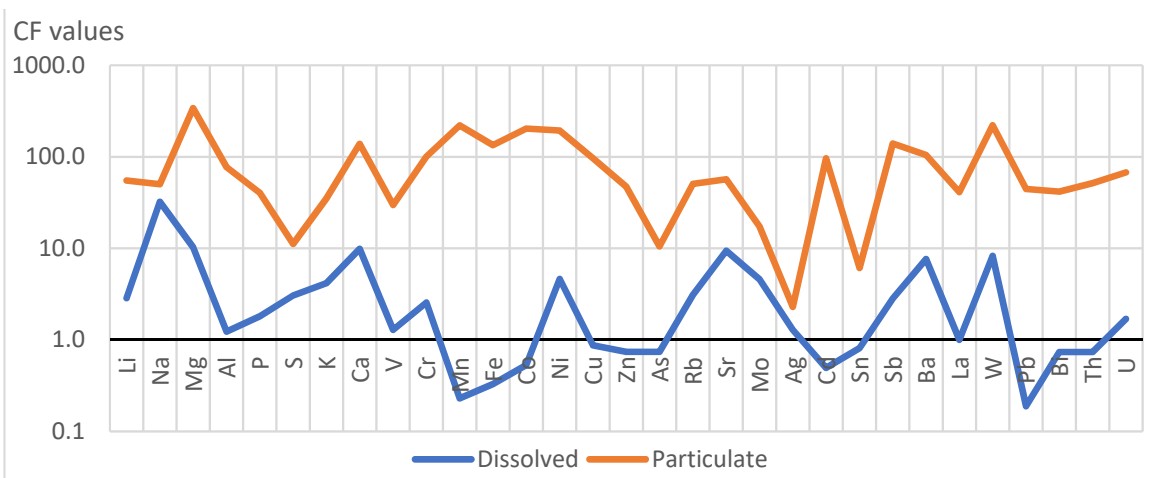

**Figure 4.** Logarithmic contamination factor values computed by using geometric means.

The EF values for trace and major elements are shown in Figure 5. The background area was characterized by the most significant enrichment in Cd, Ag, Sb, Zn and Cu (EF > 1000) and the minimal enrichment in Fe and V (Figure 5a).

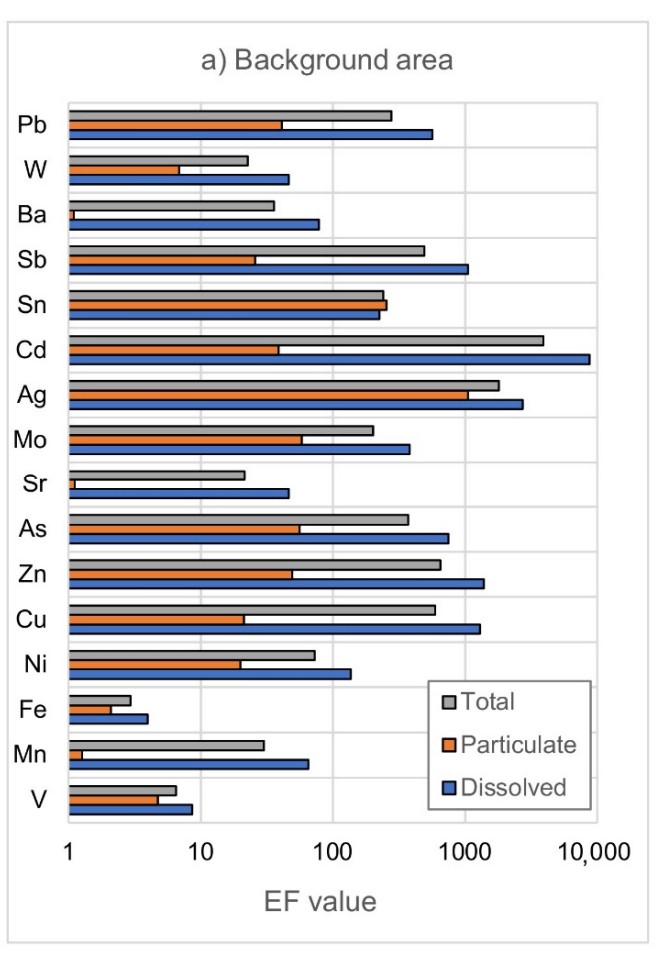

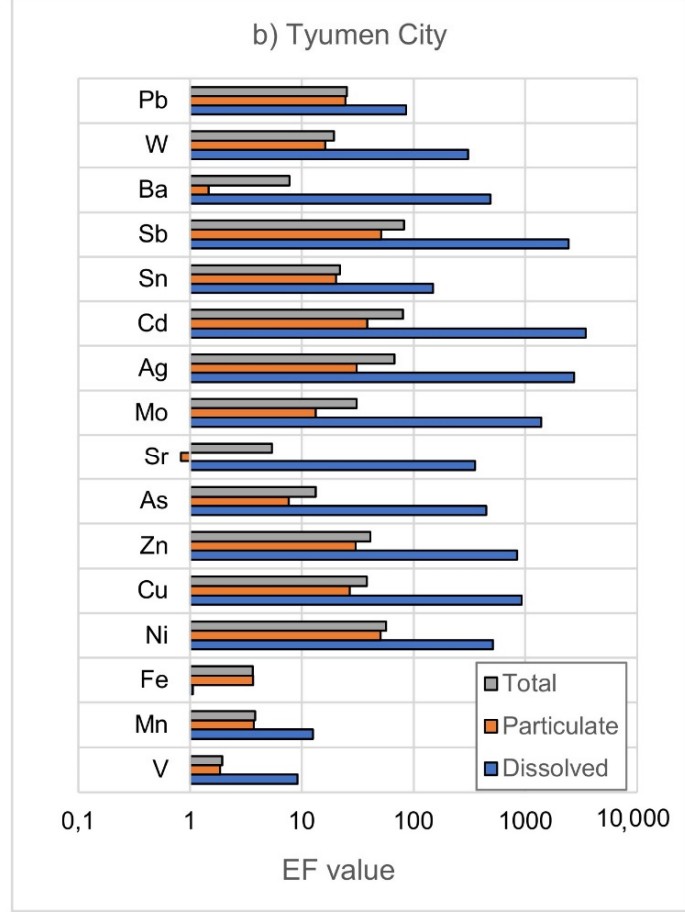

**Figure 5.** The enrichment-factor values of TMMs in the background area (**a**) and Tyumen City (**b**).

Similar results were obtained in the study by Shevchenko et al. [11] showing that snowpack within Western Siberia has a range of EF values, 1–5 (Fe) to more than 1000 (Sb, Zn and Cd). The enrichment of aerosols in Cd and Zn due to manmade pollution has been detected in many studies in different parts of the globe. Kim et al. [70] noted that Asian and non-Asian dust was strongly or extremely polluted by Cd, Zn, Pb, Cu and As. Dong et al. [71] reported that Zn and Cd were clearly enriched (EF values >100) in surface snow and a snow pit on the Northern Tibetan Plateau of China. In the snow of Teheran (Iran), maximal EF values are reported for Zn, Pb and Cd [72]. In Southern China, atmospheric aerosols are enriched in Se, Cd, Zn, Pb, As, Mo and Cu (EF > 100) [46]. The nonferrous metal industry makes a significant contribution to the emissions of Cd, Zn Pb and Cu into the environment [73]. As discussed above, high concentrations of Pb and Cd in the snowpack around Tyumen City are connected with the wind transportation of polluted air from the Urals, which is the largest metallurgical and metal processing center [66].

Based on the EF values of the TMMs deposited with the snowpack of Tyumen City (Figure 5b), we can distinguish three groups of elements as follows: (1) low enriched (EF 1–10) elements (Rb, Fe and V) sourced from soils; (2) moderately enriched (EF 10–100) elements (W, Ba, Sr, Ni, Co, Cr and Mn) that originate from mixed soil-anthropogenic sources; and (3) highly enriched elements (Pb, Sb, Cd, Ag, Mo, As, Zn and Cu) with EF values > 100, reflecting an important contribution from anthropogenic sources.

Pollution within Tyumen City was primarily associated with local sources. The main source of Pb and Sb emissions in Tyumen was the storage battery plant, around which these elements were found in highest concentrations. Until very recently this plant produced storage batteries from lead-antimony alloys containing Pb, Sb, As, Bi and Sn [74]. It is well known that vehicle emissions and road dust resuspension can be an essential source of Pb, Sb, Cu, Ni and other metals [75–77]. However, leaded gasoline has been banned in Russia since 2006. Concentrations of Pb in the street dust of Tyumen have been shown to be lower than those in the street dust of many large cities and lower than those in soils [39]. Therefore, transport has an insignificant influence on the Pb concentration in the snowpack of Tyumen, but it probably makes significant contributions to the increase in Ni, Sb and Cu concentrations.

The abrasion of tires, metal parts of cars and road markings is known as a source of Sb [76]. As noted by Hjortenkrans et al. [78], on motorway sections where traffic regularly brakes (traffic lights, intersections, etc.) and in other decelerating environments such as roads with traffic lights, roundabouts and intersections, the Sb concentration is more than 8 times the background level.

Principal sources of Cu in the atmosphere include fossil fuel burning, traffic emissions, fuel combustion and industrial combustion [79]. According to Reference [39], concentrations of Cu, Sb, As and Mo in the street dust of Tyumen City are most likely related to traffic pollution, although they can come from other sources.

Emissions of Zn and Cd originate mainly from solid waste incinerators [80]. The abrasion of automobile tires and the manufacturing and dumping of Cd batteries may be other sources of cadmium emissions [81]. Cu and Zn have a close relationship with fossil fuel combustion [82].

According to Reference [83], Ni, Fe and Cr in city dust are mainly traffic related. Diesel engines use fuels with Ni additives [84]. The most important source of Ni is the combustion of petroleum and petroleum products [80]. The global emission of Ni from the combustion of oil products is estimated to range from 10 to more than 40 kt/yr [85].

### 3.3. Contamination Levels and Risk Assessment

The values of $E_r^i$ calculated for dissolved (1), particulate (2) fractions and the total concentrations of TMM (3) calculated for the city territory are shown in Figure 6. The $E_r^i$ values of particulate forms of elements were many times higher than those of dissolved forms, which was due to the high dust deposition rates within the city that are five times higher than those in the background area. It should be noted that the CF values were also significantly higher in the particulate phase as compared to the dissolved phase (see Figure 4). Dissolved forms of all elements presented low ecological risks ($E_r^i < 40$). Particulate forms of Cd, Cu, Ni, Co, Mn, Pb, W, Sb and Cr were associated with extremely high ecological risks ($E_r^i > 320$). The mean values of $E_r^i$ indices calculated from the total content of particulate and dissolved fractions were shown to increase in the following order: Zn(5.0) < Mn(13.6) < Sn(14.4) < Mo(23.4) < Sr(23.9) < As(26.3) < V(28.8) < Sb(30.3) < Cu(30.4) < Cd(45.3) < Pb(45.4) < W(99.9) < Ni(440) < Cr(604) < Co(637). Therefore, moderate risk is associated with the atmospheric deposition of Cd and Pb; considerable risk is associated with W deposition; high risk is associated with Fe; and very high risk is associated with Ni, Co and Cr.

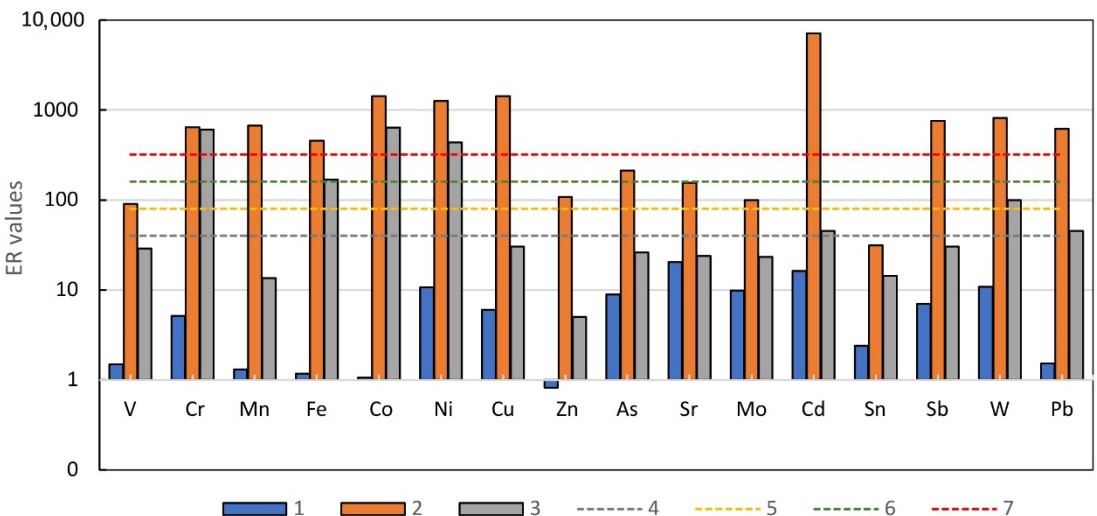

**Figure 6.** Potential ecological risk index values: 1—dissolved forms; 2—particulate forms; 3—total; 4–7—ecological risk levels: 4—low (<40); 5—moderate (40–80); 6—considerable (80–160); 7—high (320). The values above 320 characterize an extreme ecological risk.

The ranges of RI values within the Tyumen City were very wide due to the diversity of emission sources. Our calculations showed that 31% of sampling sites belong to the low risk category (RI < 80), 20%—moderate risk (80 < RI < 160), 9%—considerable risk (160 < RI < 320), and 40%—high risk (RI > 320) (Figure 7a). The distribution of RI values over different land-use areas within Tyumen showed that transport zones, industrial zones and the historical center had the most unfavorable ecological conditions with RI > 600 (high ecological risk) (Figure 7b). Near the main motorways, there were increased dust deposition rates and increased contents of particulate fractions of elements. The historical center has been affected by long-term soil pollution. Much better ecological conditions were observed in the areas of modern high-rise residential areas and the business zone that appeared between 1970 and 2020.

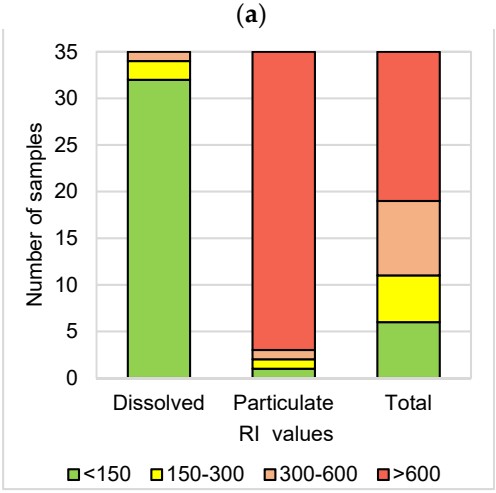
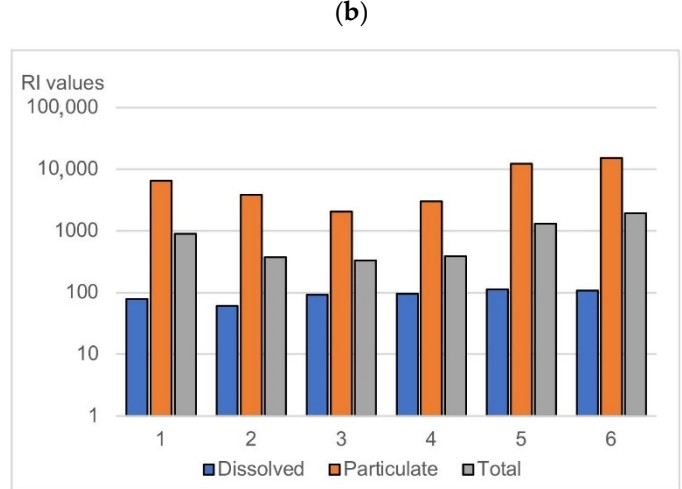

**Figure 7.** RI distribution in Tyumen City: (**a**) the frequency distribution of RI $_{total}$ values in snow samples and (**b**) mean values of RI within different land-use areas, based on the total content of particulate and dissolved TMMs in the snowpack. Land-use areas: 1—the historical center; 2—low-rise residential area; 3—high-rise residential area; 4—modern business zones; 5—industrial zones; 6—transport zones.

### 3.4. Comparison of TMMs Concentration with Other Data

In order to assess the level of pollution in Tyumen in relation to other locations, we compared the data obtained from the literature sources [22,24,57–61,67,68]. The rates of TMMs deposition within snow in different cities significantly varies depending on the local and regional anthropogenic emission sources. The range of variations in TMMs concentrations is very wide, reaching 1 to 2 orders of magnitude. Selected data from the literature are shown in Table 3.

**Table 3.** A comparison of trace element concentrations in snow from different locations ($\mu g \cdot L^{-1}$).

| City, Country, Location | Co | Ni | Cu | Zn | As | Cd | Pb | Mn | V | Reference |
|---|---|---|---|---|---|---|---|---|---|---|
| Tyumen, Russia dissolved | nd | 3.7 | 3.6 | 9.1 | 0.36 | 0.045 | 0.21 | 1.4 | 0.14 | This study |
| particulate | 1.58 | 27.8 | 8.0 | 25.0 | 0.47 | 0.038 | 4.6 | 31.5 | 2.16 | |
| total | 1.58 | 31.5 | 11.6 | 34.1 | 0.83 | 0.08 | 4.8 | 32.9 | 2.29 | |
| Moscow, traffic zones, dissolved | 0.28 | 4.6 | 5.2 | 16 | 0.05 | 0.068 | 0.35 | 11 | 0.92 | [67] |
| particulate | 1.6 | 6.1 | 1.7 | 33 | 0.27 | 0.053 | 5 | 55 | 12 | |
| total | 1.88 | 10.7 | 6.9 | 49 | 0.32 | 0.121 | 5.35 | 66 | 12.9 | |
| Vladivostok, Russia (total), city | - | 0.64 | 2.81 | 32 | - | 0.11 | 0.91 | 36.4 | 0.69 | [86] |
| suburb | - | 0.36 | 2.16 | 22.7 | - | 0.05 | 1.04 | 17.3 | 0.69 | |
| Lake Baikal and the adjacent territory(total) | 1.0 | 1.0 | 2.0 | 5.0 | - | 1.0 | 1.5 | 8.0 | 2.0 | [24] |
| Chernogolovka, Russia(total) | 4.1 | 7.8 | 5.8 | 240 | - | - | 7.2 | 141 | 16 | [87] |
| Svirsk, Russia (total) | 0.41 | 2.3 | 2.3 | 18 | 3.7 | 0.07 | 0.48 | - | 3.3 | [88] |
| Poznan, Poland (total) | - | 3.77 | 2.03 | 13.2 | 0.71 | 0.08 | 4.93 | - | - | [22] |
| Tianjin, China. 2015 year (total) | 0.17 | 1.25 | 1.96 | 22.1 | 1.37 | 0.66 | 0.17 | 13.6 | 0.34 | [89] |
| Luleo, Sweden, City Center, no-traffic area (total) | - | - | 14 | 50 | - | - | 16 | - | - | [68] |

In relation to other cities, Tyumen is characterized by a high concentration of Ni in the snow, which is only comparable with the values reported from Istanbul [11]. Concentrations of other elements are at intermediate levels and fit into the range of variations found in the other cities. For example, the Pb concentration in Tyumen is higher than those in Vladivostok [86] and Svirsk [88] (Russia), but lower than those in Chernogolovka (Russia) [87] and Luleo (Sweden) [68]. However, it should be mentioned that the latter two cities were studied in the late 1990s, when Pb additives to car fuel were widely used. The concentration of Cd, which is mainly emitted into the atmosphere through coal combustion, is quite low in Tyumen, where the thermal power stations and a significant proportion of private households use natural gas as a fuel.

## 4. Conclusions

The background sites located at distances of 25–35 km from Tyumen City were characterized by increased concentrations of soluble Cd, Cu, Zn, Pb, Ni, As and Mo as compared to their average levels in the snowpack of Western Siberia. A similar assemblage of pollutants is reported in published papers on metallurgical districts of the Southern Urals located at distances of 250–350 to the southwest of Tyumen. Due to the predominance of southern and southwestern winds in the study area during the winter, the increased concentrations of those elements can be explained by a long-range atmospheric transport from the sources located in the Urals. This conclusion is additionally substantiated by the acidification of snow due to the $SO_2$ emissions from metallurgical plants. Therefore, soluble TMM concentrations in the snowpack provide information on a long-range transport of pollutants, which can be used for modeling of the distribution of those pollutants.

The analysis of the composition of snow meltwater within Tyumen revealed significant human-induced differences as compared to the snowpack of the background areas. In particular, the contents of dissolved salts and dust in Tyumen snow meltwater were multiplied by 7 and 5 times, respectively, and pH increased from 4.7 (in the background) to 6.3 (in the city).

There was a manifold increase in TMM deposition from the atmosphere in Tyumen. The balance of dissolved and particulate fractions of elements also changed: the background area was characterized by the predominance of dissolved forms of the majority of TMMs, while the city was completely dominated by particulate forms. Therefore, a combined assessment of both particulate and dissolved fractions is necessary for a precise determination of pollution level. The TMM deposition from snow within the city was increased by 1 to 2 orders of magnitude in comparison with the background due to the increased rate of atmospheric particulate deposition and higher concentrations of the elements in dust particles. Highly enriched elements (Pb, Sb, Cd, Ag, Mo, As, Zn and Cu) with EF values >100 reflected an important input from anthropogenic sources. Calculations of RI showed that 40% of sampling sites belonged to the category of very high risk. The highest RI values were in the historical center, industrial zone and along the roads with a most intensive automobile traffic. Relatively low pollution levels were observed in the modern residential areas and business zones of Tyumen. The comparison with other cities showed that Tyumen's snowpack had a high content of Ni, while other element concentrations were at intermediate levels.

This research could offer a reference for the atmospheric pollution prevention and control in Tyumen. The data obtained in the present study allowed us to identify the most polluted parts of the city, which are located in the center and along the roads with the most intensive traffic. Ecological monitoring of Tyumen City should prioritize the assessment of particulate forms of Ni, Co Sb, Zn, Cd, Pb, Cu, As, Sn, W and Bi, which can adversely affect human health.

**Supplementary Materials:** The following are available online at https://www.mdpi.com/article/10.3390/min11070709/s1. Table S1: Description of sampling sites. Table S2: Methods of analysis, analytical results and recovery of certified reference material "Gabbro Essexit STD-2A (GSO 8670-

2005). Table S3: Methods of analysis, analytical results and recovery of certified reference material "Trace Metals in Drinking Water".

**Author Contributions:** Conceptualization and methodology, D.M.; field sample acquisition, R.P. and A.T.; writing—review and editing, D.M. and A.Z.; visualization, R.P. and A.T. All authors have read and agreed to the published version of the manuscript.

**Funding:** This research was funded by the Russian Foundation for Basic Research (project no. 19-05-50062) and project no. 121041600045-8 of RAS Siberian Branch.

**Data Availability Statement:** No applicable.

**Acknowledgments:** We would like to thank the financial support from the Russian Foundation for Basic Research (19-05-50062). The authors are especially grateful to Vasiliy Karandashev for the element determination.

**Conflicts of Interest:** The authors declare no conflict of interest.

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
