# Peer review of "Concentrations of Major and Trace Elements within the Snowpack of Tyumen, Russia"

_minerals, doi:10.3390/min11070709_

Round 1
Reviewer 1 Report
The manuscript presents interesting data on the content of dissolved and particulate fractions of chemical elements in the snow cover of the large city of Siberia (Russia) - Tyumen and the background territory, which is about 30 km away from the city against the prevailing winds. Much attention is paid to studying the ratio of the fractions of chemical elements and analyzing the solubility of elements, which has received increased attention in recent years but has not been sufficiently studied. However, there are several issues below.
Abstract: abstract lacks essential data on the ratio of the chemical elements fractions in the snowpack.More results obtained by the authors should be noted.
Line 144: “50 cm2” => “50 cm2”.
Line 147: “24 dm3” => “24 dm3”.
Line 177: “(Na2O, MgO, Al2O3, P2O5, S, K2O, CaO, Fe2O3)” => “(Na2O, MgO, Al2O3, P2O5, S, K2O, CaO, Fe2O3)”.
Line 203: Why was EF calculated relative to the upper continental crust if there is data on background levels of chemical elements in the manuscript?
Line 220: “T_r^I” – please correct.
Line 221: It is not clear why Tr = 0.5 was taken for V, Mn, Fe, Sr, W; Tr = 1 for Cr, Co, Ni, Cu, Mo, Sb; and Tr = 1.5 for Cd, As, Pb, Zn. It is generally accepted to use the following Tr values: Cr = 2, Zn = 2, Ni = 5, Pb = 5, Cu = 5, As = 10, Cd = 30, Hg = 40 (Håkanson, 1980). Please check all calculations. I think the RI values are too low.Recheck the results in section 3.3.
Lines 233-237: “Meltwaters from the background sites had an acid reaction, which is typical for snowpack within the taiga zone.Their mean pH value (4.7) was lower than the average meltwater pH (5.3) within the Khanty-Mansi Autonomous Okrug”. Such low pH values usually indicate acid precipitation (pH <5). This may indicate the emissions of acidifying agents in the city and their influence on the selected background area. Or it may demonstrate the impact of emissions from industrial enterprises in the Urals (lines 244-245). In this regard, the following statement may be incorrect: “… the locations of background sampling sites excluded any contamination of snow by pollutants from Tyumen” (lines 125-126).
Line 259: “Ib” => “Yb”?
Lines 281-292: Here are some papers on assessing the partitioning of chemical elements in snow and precipitation: 10.4236/ijg.2014.510095; 10.1007/s11270-017-3438-x; 10.3390/atmos11090907; 10.1007/s11270-019-4201-2; 10.1007/s11356-017-0576-z; 10.1016/j.scitotenv.2012.05.077; 10.1016/S0269-7491(03)00013-7; 10.1016/j.atmosres.2009.05.017; 10.1016/j.apr.2020.09.012; 10.3390/atmos11020144.
Lines 345-347: “However, it should be noted that some samples had very high concentrations (up to 144 mg· L-1) of Na, which were indicative of the use of de-icing agents.” The manuscript does not indicate which de-icing agents are used in Tyumen? Are sodium chloride or other sodium compounds used there?
Table 2: In table 2, compared to table 1, there are no data on the content of W, Pb, Bi, Th, U. Please add information on these elements.
Line 351: “Ni, Cr, Co, Ti, Pb, Sn, and Bi had less than 13% of dissolved forms” – there are no data on the content of dissolved phase of Pb and Bi in table 2.
Figure 3: It would be very interesting to see a similar figure for the solubility of chemical elements in the background samples and urban snow samples. You can easily compare the change in the partitioning of chemical elements in the city in respect with the background.
Figure 4: “Dissollved” => “Dissolved.”
Figure 4: Why did you decide to show the EF value for the dissolved chemical elements and the total content of elements but did not provide data on the EF value for the particulate elements?
Lines 394-400: It is well known that vehicle emissions and road dust resuspension can be an essential source of Sb, Cu, Pb, and other metals (e.g., 10.1007/s10661-021-09152-5; 10.1016/j.scitotenv.2018.10.214). Could the vehicles be a possible source of elements in the snow cover of Tyumen?
Figure 5, Figure 6, Figure 7: Are the data in the figures presented for the dissolved or particulate fraction of chemical elements? Or, when calculating the RI, the data for two fractions of elements were summed up?It would be informative to show the differences in the risk assessment when calculating for two fractions of elements to understand which fraction generates the highest environmental risk.
Table 3: the concentrations of which fractions are presented in the table, dissolved or particulate? It is necessary to compare the data on dissolved and particulate fractions of elements separately.
Conclusions: It is imperative to note for what purposes the obtained results can be helpful. Please add the environmental implications of the findings, even a few lines. I did not find them mentioned in the text.
The manuscript requires major revision.
Author Response
Thank you for your careful consideration of the manuscript. Corrections and answers in the attached file

Reviewer 2 Report
The paper is devoted to the highly relevant problem of studying the content of major and trace elements in suspended forms and particulate fractions in the snow-pack of Tyumen, Russia. The manuscript is well written and structured. The manuscript can be accepted in present form.
Author Response
Thank you for your careful consideration of the manuscript.
Reviewer 3 Report
The manuscript presents and discusses new data concerning the distribution of trace elements in the snowpack of Tyumen from a 2020 sampling campaign, which are useful to provide an indirect picture of air quality over an important urban area in Siberia. Overall, the study is presented clearly, coherently and with good care in supporting all statements with scientific details, elaborations, arguments, and adequate references. Trace elements data, which could be not very informative on their own, are nicely developed to extract synthetic parameters (the ecological risk index and the map in Figure 6 above all), which translate them into informative results, clear and useful for potential decision makers.
A number of (still minor overall) comments on the manuscript from my side are reported below.
General comments:
The study of elemental fractionation between dissolved and particulate forms is clearly one of the backbones of the study, but I have the feeling that the interpretative value of this aspect could be strengthened a bit more in the key sections of the manuscript (abstract, res. disc. and conclusions): why should we care about that specifically? Is it useful to discriminate the source of aerosols? Is it relevant from the ecotoxicological viewpoint? These arguments would support the reasons of an experimental design setting fractionation as an objective of the study. Otherwise, presenting it as an empirical finding to be explained a posteriori (see comparison with DTI) makes this outcome looking weak.
There is some inconsistency about numbers within the manuscript, how many samples have been analyzed precisely? 41 from the abstract (line 12), 50 (+11 background) from the methods (lines 149-150), 35 from the map (Figures 1 and 6). How many elements were determined? 60 from the abstract (line 11), 62 from the methods (line 172, is Rh maybe spiked as internal standard?), 48 from table 1, 40 from table 2.
There is some confusion along the manuscript due to the usage of both “mean” and “geometric mean”. Firstly, the former should be better reported as “arithmetic mean” for clarity. Secondly, the reason why geometric mean is used (which is a bit unusual) is not specified. Third, in section 2.4. it is not specified which one of the two is used to calculate all derived parameters. I suppose that data distributions are not normal… that should be discussed in case. However, I would suggest reporting the (nonparametric) median in place of the geometric mean, unless there are specific reasons for adopting the latter (e.g. a subsequent specific parametric statistical test for groups comparison).
The way detection limits were calculated is unclear (see specific comment), this could affect part of the results concerning low-level elements, thus I would recommend to re-check them carefully.
Even if excel is not the most professional software for graphics, I think the Figures 1,2,7 can be improved (e.g. by setting the intercept of the x-axis to the minimum of the y-axis, adding a title to the y-axis etc.).
A general (minor) review of the English is suggested, together with a general typographic revision (e.g. exponents, blank spaces, punctuation).
Specific comments:
Lines 39 to 43: the combination of these two sentences is unclear, how could impurities in snow be useful to determine soil characteristics if snow-covered soil does not significantly affect the generation of atmospheric aerosols?
Lines 57-58 yes, that is an important statement in the framework of the study settings: are there information available concerning the elemental composition of soils in the region of Tyumen?
Line 83 I would say “distribution” more than “characteristics”.
Line 88, why is fractionation important (see general comment)?
Line 93 I would immediately refer to Figure 1 as it is very useful to visualize the regional setting and land usage.
Line 119 did those short-term warmings potentially led to a (even partial or localized – site specific) melting of the snow cover?
Line 141 apart of the distance from the road, did the authors adopt other criteria, like to minimize the probability to collect the sample from a spot altered by human activities/passage (that sounds not easy to me in urban areas over the 3-4 months of time span covered by the profile)?
Figure 1 I see only 35 sampling points here, where are the others? It would be nice to visualize the location of background sites as well, maybe in an additional mid-scale map.
Line 177 why are oxides reported here (and nowhere else mentioned in the manuscript)?
Line 178 atomic
Line 181 onward analytical details are missing here: how was the acid digestion carried out? What are control samples? What is the standard sample? More than “standard samples” the two mentioned are “reference materials”. What was the typical accuracy achieved from the reference materials? If digestion was not carried out with HF I would expect lower recoveries for refractory elements (e.g. Al, Ti…).
Line 189-193 this is a very strange (and unclear, given the control sample is unspecified) way of defining the DL. It should be 3 times the SD of replicate measurements of procedural blanks. Did the authors include procedural blanks in their analysis? Was the average blank subtracted to final results? Quantification limits could be also used (10 times SD), but why is 5 times SD used as the reference here? There is no point in reporting the RSD of data <DL.
Line 239-241 again, could on-site partial/local melting before sampling explain this discrepancy?
Table 1 as a note, it is stated here that total level is the sum of the two (dissolved and particulate) geometrical means. I think it would be more correct to calculate it by summing dissolved and particulate fraction for each sample individually, and then reporting the mean. I suppose the PSE was also calculated from the ratio and sum of the geometrical means, whereas I suggest calculating it sample by sample, and then report the mean.
Section 3.2. again, some references to the geochemical composition of soils and rocks in the region (if available) would useful to support the discussion.
Figure 2 I’m not sure this plot is the best way to visualize the differences between Tyumen and regional levels… perhaps representing the ratio Tyumen/Regional would make clearer the anomalies? Also: I suppose the Tyumen data here are pooling the urban sampling site only, is it correct? Are the regional data also obtained from urban sites only?
Line 326 can this statement be supported by a reference?
Line 330 is any information about impurities in de-icing agents available? Could other trace elements come from de-icing agents’ contamination?
Table 2 Where are the elements W, Pb, Bi, Th, U? Some are discussed afterwards. Also, adjust decimals in the total column for consistency with the means.
Figure 4 correct the word dissolved.
Line 388-393 (and the abstract) I would suggest keeping in mind that those thresholds for EFs (<10;10-100;>100) are highly qualitative and conventional, thus can be used to attempt hypothesis about the origin of trace elements, not to express definite source appointment.
Section 3.3. it should be clearly pointed out that this part discusses Er calculated for the urban sampling sites only (if so).
Figure 8 is redundant with the main text, I would remove it.
Section 3.4. again, to improve comparability I would clarify which of the literature data refers to urban areas only or includes background sites as well.
Author Response

(The authors gave the same response as above.)

Reviewer 4 Report
The manuscript presented for review for the journal Minerals, entitled „Concentrations of major and trace elements within the snowpack of Tyumen, Russia”, would be very interesting for researchers worldwide. However, it would require polishing, especially in terms of manuscript structure.
Keywords: environmental monitoring – it sounds as if this was routine monitoring report and not novel research for a research article.
Introduction
Too local. It needs more global context. Please describe the general problem and only then present it upon the example of the chosen area. Please write explicitly what is the novel aspect of this work.
Material and methods
2.2 Sample collection
This subsection is too long. The data on selected locations would be better presented in a table.
2.3 Analytical procedures
The information on the applied equipment/procedure/QA/QC is best (clearest and shortest) presented in a table.
Results and Discussion
How is it justified that in Tables 1 & 2, the number of significant digits in the presented results span from one (e.g., 0.005 for Li) to five (e.g., 14.010 for Fe). The uncertainties/errors of the obtained results need to be calculated and only based on that – the actual results should be reported with an appropriate accuracy. Please correct.
The Discussion is interesting but too long, and parts of it are too simple for a research article.
Figure 7 – perhaps it would also be better to replace it with a table. There could be brackets (spans) given for particular surfaces.
Table 3 should contain the information but from areas of similar climate – e.g. other Arctic sites. Turkey is hardly comparable as background.
Conclusion
Too local.
CONCLUSION should summarize the conducted research with respect to the already published knowledge (including other regions of the world), in order to present the global impact of this work (instead of only local). The manuscript should present future prospects (what has been achieved by this investigation? who will benefit from it? what further research should be conducted?), and these should also have a global dimension.
References
Too few among the references are new, i.e. from the years 2000-2021.
Author Response

(The authors gave the same response as above.)

Round 2
Reviewer 1 Report
The authors made a comprehensive response to all comments. The only minor correction that needs to be made: in Fig. 3 (a), some elements are not visible (M... after Ca, M... after Sr, S… after Nd). It is recommended to accept the manuscript for publication.